# Fungal Jasmonate as a Novel Morphogenetic Signal for Pathogenesis

**DOI:** 10.3390/jof7090693

**Published:** 2021-08-26

**Authors:** Yingyao Liu, Martin Pagac, Fan Yang, Rajesh N. Patkar, Naweed I. Naqvi

**Affiliations:** 1Temasek Life Sciences Laboratory and Department of Biological Sciences, National University of Singapore, 1 Research Link, Singapore 117604, Singapore; Yingyao@tll.org.sg (Y.L.); yangfan@tll.org.sg (F.Y.); 2A*STAR Skin Research Labs, Singapore 138632, Singapore; martin_pagac@asrl.a-star.edu.sg; 3Department of Biosciences and Bioengineering, Indian Institute of Technology Bombay, Mumbai 400076, India

**Keywords:** cyclic AMP, fungus, jasmonic acid, LCMS, *Magnaporthe oryzae*, pathogenic development, ROS, signalling

## Abstract

A key question that has remained unanswered is how pathogenic fungi switch from vegetative growth to infection-related morphogenesis during a disease cycle. Here, we identify a fungal oxylipin analogous to the phytohormone jasmonic acid (JA), as the principal regulator of such a developmental switch to isotropic growth and pathogenicity in the rice-blast fungus *Magnaporthe oryzae*. Using specific inhibitors and mutant analyses, we determined the molecular function of intrinsic jasmonates during *M. oryzae* pathogenesis. Loss of 12-Oxo-phytodienoic Acid (OPDA) Reductase and/or consequent reduction of jasmonate biosynthesis, prolonged germ tube growth and caused delayed initiation and improper development of infection structures in *M. oryzae*, reminiscent of phenotypic defects upon impaired cyclic AMP (cAMP) signaling. Chemical- or genetic-complementation completely restored proper vegetative growth and appressoria in *opr1*Δ. Mass spectrometry-based quantification revealed increased OPDA accumulation and significantly decreased jasmonate levels in *opr1*Δ. Most interestingly, exogenous jasmonate restored proper appressorium formation in *pth11*Δ that lacks G protein/cAMP signaling; but failed to do so in the Mitogen-activated protein (MAP) kinase mutants. Epistasis analysis placed jasmonate upstream of the cAMP pathway in rice blast. Mechanistically, intrinsic jasmonate orchestrates timely cessation of the vegetative phase and induces pathogenic development via a complex regulatory interaction with the cAMP-PKA cascade and redox signaling in rice blast.

## 1. Introduction

Phytohormone mimics are produced and utilized by some pathogenic fungi in cross-kingdom communication with the hosts for immuno-modulation to aid their survival and infectivity. This striking phenotype has become a rapidly emerging research field [1,2]. We found that jasmonic acid (JA) and its derivate 12-hydroxyjasmonic acid (12OH-JA) are synthesized by the rice blast fungus *M. oryzae* and subsequently secreted into the host cells to suppress the JA-dependent plant immunity [3]. This remarkable discovery also identified that the phytopathogenic fungus *M. oryzae* follows *Lasiodiplodia theobromae* [4], *Fusarium oxysporum,* and *Gibberella fujikuroi* [5], in biosynthesis of such fungal JA-like oxygenated lipids or oxylipins in vivo.

Oxylipins are diverse secondary metabolites derived from oxidation and subsequent modification of polyunsaturated fatty acids. Fungal oxylipins, like their mammalian counterparts, possess hormone-like functions and play key roles in both asexual development and pathogenesis [6]. Recent studies highlight the interesting functions of fungal oxylipins, particularly at the host-pathogen interface [3,6]. Accumulation of different oxylipins, in the host environment, influences the morphology and subsequent biofilm formation by the opportunistic human fungal pathogen *C. albicans* [7,8,9,10]. For example, prostaglandins, the mammalian oxylipins with different hormone-like functions (including immune response), are found to induce the yeast to hyphal transition in *C. albicans* [8]. Intriguingly, *C. albicans* can produce PGE2—one such prostaglandin—in the presence of exogenously provided polyunsaturated fatty acid arachidonic acid [11]. On the other hand, antifungal properties of polyphenolic phytoalexins of plant hosts are likely associated with decreased fungal oxylipin production, leading to altered morphogenesis and subsequent reduced virulence in the pathogen. For example, phytoalexin nordihydroguaiaretic acid (NDGA) has been shown to induce morphogenetic switch over from the pathogenic hyphal form to the non-pathogenic yeast form in the Dutch elm pathogen *Ceratocystis ulmi* [12]. While the underlying mechanism is still not clear, it is believed that the oxylipins may be involved in such morphological changes, especially given the inhibitory effect of these phytoalexins on fungal fatty acid oxidases and prostaglandin production in *Cryptococcus neoformans* [13].

The biosynthesis and physiological functions of jasmonate have been well studied in plants. Briefly, polyunsaturated fatty acids, such as α-LeA, are synthesized from a membrane component and secreted into plastids [14]. Sequential catalysis by 13-Lipoxygenases (13-Lox), Allene oxide synthase (Aos), and Allene oxide cyclase (Aoc) [15,16,17,18,19,20,21], generates the key intermediate product, *cis*-OPDA, in the chloroplasts [18,19,21,22,23]. Upon the migration of *cis*-OPDA to peroxisomes, it is reduced by Opr3, and then its carboxylic acid side chain is shortened via three cycles of β-oxidation reactions [24,25]. Finally, *cis*-jasmonic acid is biosynthesized in the peroxisomes, and trafficked to the cytosol where it can be modified into different kinds of jasmonates [26]. Although the mechanism remains unclear, recent studies suggest that jasmonate and/or its derivates are also secreted out of the cells [3].

Compared to the knowledge in plants, the synthesis and metabolism of jasmonate-like oxylipins in fungi remain largely unexplored. For instance, OPDA, the critical intermediate in JA biosynthesis, has been identified in *L. theobromae* and *F. oxysporum*. However, the relevant metabolic enzymes, such as Allene oxide synthase (Aos) and OPDA reductase (Opr), and the biosynthesis pathway have not been characterized in detail [27,28,29]. Another central question that has remained unexplored is: What are the physiological functions of oxylipins or jasmonate(s) in fungi? Thus far, the investigation of fungal jasmonate function has predominantly focused on modulation of the plant immune response. For example, the most well-elucidated microbial jasmonate is Coronatine (COR) from the pathogenic bacterium *Pseudomonas syringae* [30]. COR serves as a molecular mimic of the bioactive JA-Isoleucine and competes with binding to the co-repressor Coi1, which activates JA signalling and suppresses salicylic acid-based plant defence [31]. Likewise, Lasiojasmonate (LasA), an inactive jasmonate derivative produced by *L. mediterranea*, stimulates JA-Ile production at a later stage of infection to activate the cell death response and JA signalling in the host plants [32]. Nevertheless, the physiological role of such intrinsic JA and its derivatives remains unclear in these fungal systems.

The Blast pathosystem, comprising Rice and *M. oryzae*, represents one of the most destructive fungal diseases of cultivated cereal crops. *M. oryzae* is a hemibiotroph that initially keeps the host cells alive prior to switching on the destructive necrotrophic cell death modality therein [33]. Conidia/asexual spores of the blast fungus follow an intricate developmental program in response to specific surface cues to differentiate into specialized infection structures, the appressoria, at the tips of emerging germ tubes [34] (Figure 1a). In the absence of such inductive surface cues, the germ tubes continue vegetative development in a unipolar fashion. An enormous turgor pressure generated within the appressorium is utilized to physically rupture the host cuticle, and a thin penetration peg enables entry into the host [34]. The peg further differentiates into bulbous infectious hyphae that colonize host tissues, resulting in typical blast symptoms.

Here, we demonstrate that the phytopathogenic fungus *M. oryzae* synthesizes jasmonic acid via a fungal OPDA reductase, MoOpr1. Interestingly, loss of *OPR1* significantly reduced the overall JA levels and increased *cis*-OPDA accumulation in vivo. Our study revealed that JA synthesis in *M. oryzae* follows the octadecatrienoic pathway akin to plants, with *cis*-OPDA as the key precursor moiety. We identify an essential function for fungal jasmonate and *cis*-OPDA in appressorium development and function in rice blast. Our findings reveal a hitherto unknown JA signalling circuit that functions intricately with the G protein cascade as a chemico-morphogenetic switch that enables proper pathogenic development and virulence in the devastating rice blast fungus.

## 2. Materials and Methods

### 2.1. Fungal Strains, Growth Conditions and Genetic Transformation

*Magnaporthe oryzae* strain B157, obtained from the Indian Institute of Rice Research (Hyderabad, India), served as the wild type in this study. For growth and conidiation, wild type and transformants were cultured on prune agar medium (PA; per liter: 40 mL of prune juice, 5 g lactose, 5 g sucrose, 1 g yeast extract, and 20 g agar, pH adjusted to 6.5 with NaOH) or complete medium (CM; per liter: 6 g yeast extract, 6 g casein hydrolysate, and 10 g sucrose and 20 g agar). Assessment of colony characteristics and vegetative growth was carried out by growing the indicated strains in PA or CM medium at 28 °C for 3–5 days. For conidiation, fungal strains were cultured in the dark at 28 °C for 3 days, followed by continuous light for an extra 5 days at room temperature. Mycelia used for genomic DNA and protein extractions were 2–3 days’ old cultures harvested from liquid CM at 28 °C.

*Agrobacterium tumefaciens*-mediated transformation (ATMT) was applied to generate fungal transformants. BM (Basal medium; 1.6 g yeast nitrogen base without amino acid and ammonium sulfate, 2 g asparagine, 1 g ammonium sulfate, and 10 g glucose and 20 g agar, pH adjusted to 6.0 with Na_2_HPO_4_) containing 40 mg/mL ammonium glufosinate or chlorimuron ethyl (sulfonylurea, Cluzeau Info Labo, Sainte-Foy-la-Grande, France), and CM with 250 mg/mL hygromycin (A.G. Scientific Inc., San Diego, CA, USA) was used to select the indicated transformants. *A. tumefaciens* AGL1 was used for T-DNA insertional transformations. Requisite transformants were screened by Southern blot analysis and/or locus-specific PCR, and two confirmed strains selected in each instance for further investigation. *Escherichia coli* strain XL1 was used for bacterial transformations, and maintenance of various plasmids, using the specified methods as described [35].

### 2.2. Nucleic Acid Manipulations

Homology search of DNA/protein sequences was carried out using NCBI BLAST [36]. Multiple protein sequences alignments were performed with ClustalW [37], and rendered using BoxShade, Available online: https://embnet.vital-it.ch/software/BOX_form.html (accessed on 4 May 2021). Oligonucleotide primers used in this study are listed in Appendix A. Plasmid DNA was extracted from *E. coli* with Geneaid high-speed plasmids mini kit. Fungal genomic acid DNA was isolated with the MasterPure^TM^ Yeast DNA Purification Kit (Lucigen, WI, USA) following the manufacturer’s instructions. Nucleotide sequences were analyzed by the ABI Prism Big Dye Terminator Method (PE Applied Biosystems, Waltham, MA, USA).

### 2.3. Generation of Constructs for Gene Deletions or Epitope Tagging

For gene deletion of *OPR1*, DNA fragments (~1 kb) of 5′ UTR and 3′ UTR were PCR amplified, digested, and ligated sequentially to flank the *BAR* resistance cassette in pFGL822-2. GFP-Opr1 construct, the N terminal tagging with the native promoter, was created by fusing GFP fragment with *OPR1* ORF at its C terminal and downstream of its 1kb promoter, sequentially cloned into pFGL1010. This plasmid had a sulfonyl urea resistance and contributed to ectopic single-copy integration by introducing the gene into the ILV2 locus [38]. Primers used in the plasmid construction were listed in Appendix A. The gene deletion or epitope tagging constructs were introduced into *M. oryzae* B157 via *Agrobacterium-*mediated transformation to replace the target genes specifically.

### 2.4. Appressorial Assays and Chemical Complementation Analyses

For appressorial assays, conidia were collected from a 7-day old PA plate by scraping with an inoculation loop in the presence of sterile water. The conidial suspension was obtained by filtering through two layers of Miracloth (Calbiochem, San Diego, CA, USA), centrifuging and re-suspending in sterile water at a required concentration (106 conidia per mL). Droplets (~20 μL) of conidial suspension were placed on the indicated surfaces and incubated under humid conditions at room temperature. Appressorium initiation/formation was observed and quantified at 4 h, 24 h, and 28 h post inoculation (hpi), denoted in the legends. Restoration of appressoria formation was carried out by adding Jasmonic acid (JA, Sigma-Aldrich, Darmstadt, Germany), phenidone (PHEN, Sigma-Aldrich, Darmstadt, Germany), 12-oxo Phytodienoic Acid (*cis*-OPDA, Cayman, Ann Arbor, MI, USA), Methyl jasmonate (MeJA, Sigma-Aldrich, Darmstadt, Germany), α-Linoleic acid (α-LA, Sigma-Aldrich, Darmstadt, Germany), Sig8-bromoadenosine 3′,5′-cyclic monophosphate sodium salt (8-Br-cAMP, Sigma-Aldrich, Darmstadt, Germany) at 0 hpi to a final concentration of 0.2 mM, 0.3 mM, 10 μM, 0.5 mM, 0.5 mM, and 10 mM, respectively [34,39].

### 2.5. Protein Isolation and Western Blot Analysis

Total protein was extracted from 2 to 3 day old CM cultured mycelia, which were ground to fine powder with liquid nitrogen, and re-suspended in protein extraction buffer (10 mM Tris-Cl, pH 7.5; 150 mM NaCl; 50 mM NaF; 0.5% NP40; 0.5 mM EDTA; 1 mM PMSF and 1X Protease Inhibitor Cocktail) [39]. About 20 mg total proteins in each sample were fractionated by 10% SDS PAGE gel and transferred to PVDF membranes (Millipore Corporation, Burlington, MA, USA) and immunoblotted with an anti-GFP antibody (Invitrogen-Molecular Probes, Thermo Fisher Scientific, Waltham, MA, USA). IRDye680-conjugated anti-rabbit secondary antibodies were used to detect the target proteins by Odyssey Infrared Imaging System (LI-COR, Lincoln, Dearborn, MI, USA). Anti-MAPK and anti-TePY (Cell Signaling Technology, Beverly, MA, USA) were used as primary antibodies following the manufacturer’s instructions. Coomassie blue staining served as a loading control in this study.

### 2.6. Chemical Analysis (Liquid Chromatography-Mass Spectrometry)

Germlings were harvested after growth for 48 h in liquid Complete Medium and washed thrice with pre-chilled PBS. The recovered fungal biomass was then frozen immediately at −80 °C and lyophilized and ground to a fine powder using a mortar pestle and liquid nitrogen. This powdered biomass was immediately stored at −80 °C until further use. 

Lipids were extracted from 100 mg of ground germlings overnight at 4 °C with 4.5 mL of methanol (MeOH): water (H_2_O) (25:75 by *v*/*v*) at 750 rpm (revolution per minute) in a thermomixer (Eppendorf, Hamburg, Germany). Lipids from 100 µL of conditioned media were extracted by vortexing for 2 min with 4.5 mL of methanol (MeOH): water (H_2_O) (25:75 by *v*/*v*).

Extracts were centrifuged at 4 °C for 10 min at 4000 rpm in an Eppendorf 5810R centrifuge, and supernatant transferred into new 15 mL Eppendorf tubes, and topped up to a final volume of 6 mL with methanol (MeOH): water (H_2_O) (25:75 by *v*/*v*).

Samples were spiked with 50 μL of a deuterated internal standard solution containing PGD2-d4 and PGE2-d4. Analytes were extracted using Strata-X 33 mm Polymeric solid reversed phase (SPE) extraction columns (8E-S100-TGB, Phenomenex). Columns were conditioned with 3 mL of 100% MeOH and then equilibrated with 3 mL of H_2_O. After loading the sample, the columns were washed with H_2_O: MeOH (75:25 by *v*/*v*) to remove impurities, and the metabolites were then eluted with 1.5 of 100 % MeOH. The eluant was dried under vacuum and resuspended in 70 μL of ACN/water/formic acid (10/90/0.1, *v*/*v*). The extracted samples were then subjected to mass spectrometry analysis.

High-performance liquid chromatography (HPLC) coupled to triple-quad mass analysis was performed by using the Shimadzu LCMS-8060 system. Reversed phase separation was performed on a Phenomenex, Kinetex C8 (2.1 × 100 mm I.D × 150 mm L., 2.6 µm) column and maintained at 40 °C. The mobile phase consisted of (A) water/formic acid (100/0.1, *v*/*v*) and (B) ACN. The stepwise gradient conditions were carried out for 30 min as follows: 0 min, 10% of solvent B; 0–5 min, 10–25% of solvent B; 5-10 min, 25–35% of solvent B; 10–20 min, 35–75% of solvent B; 20–20.1 min, 75–98% of solvent B; 20.1–28 min, 98% of solvent B; 28–28.1 min, 98–10% of solvent B; and final 28.1–30 min, 10% of solvent B. The flow rate was 0.4 mL/min, injection volume was 10 µL, and all samples were maintained at 4 °C throughout the analysis. 

A mixture of representative native and internal standards was injected and run with the column to optimize the source parameters. The electrospray ionization was conducted in positive mode. Drying gas temperature was set at 270 °C with a gas flow of 10 L/min. Sheet gas temperature was set at 250 °C with a gas flow of 10 L/min. The nebulizer gas flow was 230 kPa. The dynamic MRM option was used and performed for all compounds with optimized transitions and collision energies. MRM transitions (precursor and product ions) and collision voltages were as follows: *cis*-OPDA (293 → 81.15; −30 eV), Jasmonic acid (211.1 → 133.15; −13 eV). The determination and integration of all peaks was manually performed using the LabSolutions Insight software. Peaks were smoothed before integration and peak to peak Signal/Noise ratios were determined using the area under the peaks.

### 2.7. Nitro Blue Tetrazolium Staining

Conidial suspension was inoculated on artificial hydrophobic coverslips with or without the indicated chemicals. At 4 hpi, the droplets were removed and replaced with an equal volume of 0.1% Nitro blue tetrazolium (NBT; Sigma-Aldrich, Darmstadt, Germany) solution, staining for 20 min. The staining was stopped using 100% ethanol, and the samples washed with water three times before the imaging process.

### 2.8. Live Cell Microscopy and Image Analyses

Time-lapse or live-cell fluorescence microscopy was performed using a Zeiss Axiovert 200 M microscope (Plan Apochromat 1006, 1.4NA objective) with an Ultra-View RS-3 spinning disk confocal system (PerkinElmer Inc., Shelton, CT, USA) which was equipped with a CSU21 confocal optical scanner, 12-bit digital cooled Hamamatsu Orca-ER camera (OPELCO, Sterling, VA, USA) and a 491 nm 100 mW and a 561 nm 50 mW laser illumination under the control of MetaMorph Premier Software (Universal Imaging, New York, NY, USA) [39,40]. Briefly, z-stacks comprised of 0.5 µm-spaced sections that were captured. GFP excitation were performed at 491 nm (Emission 525/40 nm).

Bright-field microscopy was carried out by an Olympus IX71 microscope (Olympus, Tokyo, Japan) equipped with a Plan APO 100X/1.45 objective. Images were captured by Photometrics CoolSNAP HQ camera (Tucson, AZ, USA) and processed with MetaVue (Universal Imaging, Downingtown, PA, USA), Adobe Illustrator (Adobe Inc., San Jose, CA, USA), and ImageJ (LOCI, University of Wisconsin, Madison, WI, USA).

### 2.9. Plant Cultivar and Pathogenicity Assays

Rice cultivar CO39 susceptible to *M. oryzae* strain B157 was utilized for pathogenicity assays. Rice seeds were soaked in water and placed at room temperature for 5 days to germinate. The rice seedlings were grown at 80% humidity, 28 °C in a 16-h light (28 °C)/8-h dark (22 °C) cycle, for 2 weeks. For blast infection assays, fresh conidial suspension (1 × 10^6^ conidia/mL with 0.01% gelatin) was sprayed on the rice seedlings, and inoculated seedlings were grown for an extra 7 days in a growth chamber (22 °C, 80–95% humidity, 16 h illumination per day), and the assessment of blast disease symptoms was conducted as described [41].

## 3. Results

### 3.1. Fungal Jasmonate Is Essential for Timely Cessation of Germ Tube Growth during Pathogenic Differentiation in M. oryzae

Based on the JA biosynthesis pathway and sequence similarity to the orthologous gene (AT2G06050) in Arabidopsis, we predicted *OPR1* (12-oxophytodienoate reductase 1, *MGG_10583*; UniProt accession G5EHQ2; 44.2% similarity to AtOpr1; Appendix A) as a candidate locus for intrinsic jasmonate synthesis in the rice-blast fungus *M. oryzae*. To investigate the cellular function, an *opr1*Δ strain was generated by replacing the entire *MGG_10583 ORF* with bialaphos-resistance marker cassette in the *M. oryzae* wild-type strain; and a genetically complemented strain was also created by introducing the full-length *GFP-OPR1* into the *opr1*Δ (Appendix A). A vast majority of *opr1*Δ conidia produced significantly longer germ tubes compared to the WT or the genetically complemented *opr1*Δ strain (Figure 1b) under inductive in vitro (cover glass) and in planta (rice sheath) conditions. Detailed quantification and statistical analyses showed that the average germ tube length in WT and *opr1*Δ strain was 9.8 and 16.0 micron, respectively (Figure 1c), at the time of infection structure formation. Such defects or delay in pathogenic differentiation could be completely suppressed (germ tube length 10.0 micron) by native expression of GFP-Opr1 in the *opr1*Δ mutant (Figure 1b,c).

In plants, loss of *OPR* function leads to deficiency in JA production because of a failure to catalyze *cis*-OPDA to 3-oxo-2(2′[*Z*]-pentenyl)cyclopentane-1-octanoic acid, and the resultant defects can be suppressed by exogenous JA but not the precursor *cis*-OPDA [22,42]. Interestingly, chemical complementation with exogenous jasmonic acid was sufficient to restore proper germ tube development prior to initiation of infection structures in the *opr1*Δ mutant (Figure 2a,b). On the other hand, such additional JA dramatically shortened the germ tubes in a dose-dependent manner in the wild-type strain. Conversely, treatment of WT *M. oryzae* with phenidone, an inhibitor of JA biosynthesis, resulted in significantly elongated germ tubes reminiscent of the *opr1*Δ phenotypic defect of delayed appressorium formation (Figure 2c,d). Taken together, we conclude that intrinsic jasmonic acid is essential for a proper and timely switch-over from vegetative to pathogenic development (appressorium formation) in *M. oryzae*; and based on chemical complementation and inhibitor analyses further infer that *OPR1* is involved in JA biosynthesis in the blast pathogen.

### 3.2. Opr1 Is Required for JA Biosynthesis in M. oryzae

Based on the chemical complementation experiment, we posited that JA synthesis and/or accumulation was significantly reduced/altered in the *opr1*Δ strain. To investigate this further, we decided to utilize LC-MS to quantify the levels of JA and its major precursor *cis*-OPDA in the presence or absence of Opr1 function in *M. oryzae*. Targeted metabolite profiling demonstrated that about 34.85 pg/mL fungal JA (*m/z* 211.10) was produced by the wild-type *M. oryzae* strain. Loss of Opr1 nearly halved the overall JA accumulation in the *opr1*Δ mutant (Figure 3a). In contrast, the substrate of Opr1, *cis*-OPDA (*m/z* 293.00), showed a markedly increased accumulation in the *opr1*Δ mutant, which was estimated to be about five-fold higher than the WT levels (Figure 3b). Based on such excessive accumulation of *cis*-OPDA and significantly reduced JA levels in the *opr1*Δ mutant, we construe that Opr1 utilizes OPDA as a substrate and plays an important role in jasmonate/oxylipin biosynthesis in the rice blast fungus.

### 3.3. Subcellular Localization of GFP-Opr1 in M. oryzae

An *opr1*Δ strain expressing GFP-Opr1 was generated by introducing an ectopic full-length native *OPR1* locus fused in-frame with GFP at the N terminus to analyze the subcellular localization during asexual and pathogenic development in *M. oryzae* (Appendix A). The introduction of such ectopic *GFP-OPR1* under native regulation restored normal germ tube growth in the *opr1*Δ (Figure 1b,c), indicative of GFP-Opr1 being fully functional in vivo. The GFP-Opr1 signal was found to be uniformly distributed in the cytosol in mycelia and aerial hyphae (Appendix A). Interestingly, the cytosolic GFP-Opr1 congregated as punctate/vesicular structures in the terminal cell of the conidium, which subsequently formed the germ tube (Figure 3c). Such GFP-Opr1 punctae or vesicles eventually migrated and distributed along the growing germ tube. Upon appressorium formation (~8 hpi), the punctate GFP-Opr1 signal diminished and the uniform cytosolic distribution, similar to that observed in mycelia or aerial hyphae, became more prominent. At 24 hpi, the GFP-Opr1 was undetectable in the conidium but appeared as punctate structures in the nascent and mature appressorium (Figure 3c). During host penetration, the mature appressoria form penetrate pegs to initiate the invasive growth. We found that the subcellular localization of GFP-Opr1 at this in planta stage was cytosolic—similar to that observed in vegetative hyphae (Appendix A). We infer that the punctate or vesicular pool of GFP-Opr1, evident during germination and the early stages of appressorium formation, is indicative of the pivotal role of Opr1/JA in the precise regulation of germ tube growth concomitant with pathogenic differentiation in the rice blast fungus.

### 3.4. Opr1/JA Signaling Pathway Is Essential for Proper Initiation of Appressorium Formation in M. oryzae

Based on the subcellular localization pattern of GFP-Opr1 and the aforementioned characterization of *opr1*Δ, we next asked whether fungal jasmonate plays a role in appressorium formation per se in addition to its function in precise cessation of germ tube development. At 4 hpi, 58.2 ± 6.1% WT conidia initiated appressorium formation, whereas only 34.7 ± 1.4% conidia in *opr1*Δ showed the typical hooking at the germ tube tips. Thus, at this early time point, >60% of *opr1*Δ conidia failed to initiate the development of an infection structure (Figure 4a,b). Exogenous addition of JA caused a significant increase in the ability of such *opr1*Δ conidia to initiate appressorium formation and advanced the time for such induction of pathogenic differentiation to that observed in wild type *M. oryzae* (Figure 4a–c).

In order to uncouple such dual function of JA during pathogenic development i.e., (i) cessation of vegetative growth (ii) initiation of appressorium formation, we tested the effect of JA on WT *M. oryzae* conidia under a non-inducive condition (soft hydrophilic surface) that does not promote appressorium formation. Under such non-inductive conditions, exogenous JA was found to impede and/or inhibit vegetative growth in germ tubes in a dose-dependent manner (Figure 5a). We conclude that Jasmonic acid acts as a signalling molecule essential for modulation of the vegetative-to-pathogenic switch involved in first arresting the elongation of germ tube and then promoting appressorium initiation in a precise and timely manner in the rice blast fungus.

### 3.5. Functional Dependency and Crosstalk between JA and Cyclic AMP Signalling during Pathogenic Development in M. oryzae 

The abnormally elongated germ tubes are a hallmark of mutants defective in cyclic AMP signalling, leading to a delay in or loss of appressorium formation in *M. oryzae* [33,43,44]. Therefore, we reasoned that *JA/*Opr1 function likely cooperates/intersects with the cAMP signalling cascade during appressorium initiation in *M. oryzae*. We hypothesized that a crosstalk might occur between JA or related oxylipin(s) and the canonical cAMP/MAP kinase signalling, which plays a pivotal role in appressorium development in *M. oryzae* (Figure 5b). Exogenous cAMP was added to the *opr1*Δ conidia and inoculated on the inductive surface to assess the effect on germ tubes and appressorial development. Quantitative analyses at 24 hpi revealed that exogenous cAMP shortened the germ tube length from 17.3 μm to 14.0 μm (*** *p* = 1.3 × 10^−10^, two-tailed *t*-test) (Figure 5c,d) in the *opr1*Δ mutant. This led us to conclude that cAMP is able to compensate for the loss of JA to some extent; and further infers that a potential functional crosstalk likely occurs between JA and cAMP pathways during the process of appressorium development in rice blast.

As depicted in Figure 5b, Pth11 is a bona fide G-protein coupled receptor (GPCR) that anchors the G-protein/cAMP signalling in *Magnaporthe* [34,39,45]. Loss of *PTH11* results in a complete loss of cAMP signalling, leading to highly extended germ tubes that fail to form appressoria. Here, we decided to test whether exogenous JA or the related oxylipins could suppress such *pth11*Δ defects of such elongated germ tubes and lack of appressorium formation. Remarkably, JA or *cis*-OPDA (0.2 mM and 10 μm, respectively) could significantly shorten germ tube length and advance the induction of appressorium formation in the *pth11*Δ mutant (1.9 ± 1.2% (untreated) to 60.6 ± 8.2% and 22.5 ± 2.9%, respectively) (Figure 6a,b). Furthermore, methyl-JA, another oxylipin derived from JA, also restored appressorium formation in *pth11*Δ to 26.7 ± 1.7% (Figure 6a,b), which was similar to OPDA treatment. In contrast, α-Linolenic acid (α-LeA), the substrate of Lox1 in JA biosynthesis, failed to trigger appressorium formation in *pth11*Δ (Figure 6a,b). To sum up, JA and its derivative (MeJA) and precursor (*cis*-OPDA) but not α-LeA can restore proper germ tube development and induce timely appressorium formation in *pth11*Δ, showing that these exogenous JA-related oxylipins serve a similar function downstream of Pth11 in the cAMP pathway to regulate appressorium formation in *M. oryzae*, with JA being (comparatively) the most effective oxylipin (Figure 6a) therein.

Next, we checked the host penetration ability of JA-triggered appressoria in *pth11*Δ on rice sheath. About 47 ± 11% such appressoria penetrated the host cells and formed invasive hyphae at 28 hpi, indicating that exogenous JA-induced appressoria were, indeed, functional (Figure 6c,d). Taken together, we infer that a potential crosstalk occurs between JA and cAMP signalling during pathogenic development, and that the functions of these two signalling moieties therein could be reciprocally substituted in *M. oryzae*.

CpkA, the cAMP-dependent protein kinase, functions downstream of the Pth11/G protein signalling (Figure 5b), and the typical phenotypic defects in proper germ tube growth and appressorium initiation in *cpkA*Δ are similar to those observed in *opr1*Δ (Figure 7a). Therefore, to understand the mechanistic basis of the crosstalk between JA and cAMP, we asked whether JA or *cis*-OPDA could restore proper appressorium development in *cpkA*Δ too. Under normal conditions, about 58.2 ± 6.1% conidia from the WT initiated appressorium formation with a hooked structure at the tip of the germ tubes at 4 hpi; whereas only 16.32 ± 2.3% of *cpkA*Δ conidia were able to do so at this time point (Figure 7a,b). Interestingly, exogenous JA remarkably suppressed this delay, and increased appressorium initiation percentage to 35.8 ± 3.3% in the *cpkA*Δ conidia (*p* = 0.004, ** *p* < 0.01, two-tailed *t*-test) (Figure 7a,b and Appendix A). However, only 6.5 ± 1.1% conidia initiated appressoria in the *cis*-OPDA-treated *cpkA*Δ (*p* = 0.014, * *p* < 0.05, based on two-tailed unpaired *t*-test). Such OPDA-treated germ tubes were short, but most of them did not possess a hooked structure at the tip at this early stage (Appendix A and Figure 7b).

Further continued (24 h) treatment with JA or *cis*-OPDA still showed impaired appressorium initiation and formation in the *cpkA*Δ conidia. Although the germ tubes were short, the percentage of appressorium initiation in such JA or *cis*-OPDA treated *cpkA*Δ conidia was significantly reduced from 78.1 ± 9.5% to 49.1 ± 8.4% and 43 ± 6.4%, respectively (Appendix A and Figure 7c). Neither JA nor cis-OPDA suppressed the overall morphological defects in *cpkA*Δ appressoria at 24 hpi. Instead, the percentage of appressoria formed was reduced in the oxylipin treated group, and a significant number of appressoria in such JA- or OPDA-treated *cpkA*Δ samples were malformed, aberrant, or smaller in size compared to the untreated mutant or the WT control groups (Figure 7c and Appendix A). Likewise, OPDA treatment in WT led to elongated germ tubes and such abnormalities in appressorium formation (Figure 7c and Appendix A). Furthermore, the overall CPKA enzyme activity was, surprisingly, unperturbed in the *opr1*Δ, and the JA-treated or nascent *pth11*Δ mutant strains (Figure 7d) and was comparable to the WT levels at these time points. Likewise, the levels of active/phosphorylated Pmk1 MAP kinase (MAPK essential for appressorium formation) remained unaltered in the *opr1*Δ mutant (Appendix A); and exogenous JA failed to restore appressorium formation in the *pmk1*Δ and the *mst11*Δ (MAPKKK) or *mst7*Δ (MAPKK) mutant (Appendix A).

We conclude that although JA and *cis*-OPDA are able to cause timely cessation of germ tube growth, these two oxylipins are incapable of restoring proper appressorium formation in the *cpkA*Δ, thus indicating a plausible complex role for CpkA as a downstream effector in JA signalling in *M. oryzae*. We also infer *cis*-OPDA to be an independent signaling moiety; and that the ability of exogenous JA (or OPDA) to suppress the *pth11*Δ or *opr1*Δ defects does not involve the utilization and/or induction of CPKA or MAPK activity therein. 

### 3.6. Intrinsic Jasmonic Acid Modulates Redox Signaling in M. oryzae

Previously, it has been reported that the reactive oxygen species (ROS)/redox signaling is important for proper appressorium development in *M. oryzae* [34,46,47]. Furthermore, such ROS accumulation is extremely fine-tuned and differentially regulated during germ tube growth and appressorium formation [34]. Having ruled out the engagement of MAPK and CPKA activities per se as direct downstream effectors of JA function in precise regulation of germ tube growth, we next asked whether the aforementioned ROS levels and/or redox signaling is instrumental in its physiological function(s), since such oxidant responses have recently been shown to be important for cAMP-dependent pathogenic differentiation in *M. oryzae* [34]. We assessed and quantified the overall ROS levels in nascent or JA-treated *opr1*Δ and *pth11*Δ by NBT staining. The accumulation of ROS in the germ tube tips was significantly lower in *opr1*Δ, *pth11*Δ, and *cpkA*Δ compared to the wild-type *M. oryzae* (Appendix A). Interestingly, treatment with exogenous JA significantly induced the accumulation of such ROS in the germ tubes termini (Appendix A) in the aforementioned three mutant backgrounds. Furthermore, co-treatment of *opr1*Δ or *pth11*Δ with ascorbic acid (a general antioxidant) negated the effect of exogenous JA and failed to shorten the germ tube length and/or initiate appressorium formation therein, similar to the untreated mutant strains (Appendix A). Taken together, we infer that a likely mode-of-action for JA-based suppression of germ-tube defects in the *opr1*Δ and *pth11*Δ is to induce and/or buffer the redox homeostasis in these mutant cell types that were found deficient in the overall oxidant capacity. Lastly, we construe that JA signalling cooperates with the cAMP-PKA cascade to determine the cellular redox state during specific stages of infection-related morphogenesis and plays a key role in modulating the vegetative-to-pathogenic switch in the rice blast fungus.

## 4. Discussion

We describe a fungal enzyme, Opr1, involved in JA biosynthesis in the rice-blast pathogen *M. oryzae*. Our findings show that the MoOpr1 (*MGG_10583*) belongs to the OPR-II group and is responsible for conversion of 12-OPDA, suggesting that JA biosynthesis in *M. oryzae* likely occurs via a pathway similar to that in plants. A previous study [48] indicated that *Fusarium oxysporum f.* sp. *tulipae* synthesizes JAs via allene oxide and 12-OPDA as intermediates too.

Several isozymes of OPDA reductase, with differential substrate stereospecificity, have been identified in various plant species—5 in *Arabidopsis*, 13 in rice and 3 in tomato [49]. The OPDA reductase-based conversion of cis-OPDA occurs in peroxisomes in plants [50]. In silico analysis showed that *M. oryzae* produces at least two isoforms of OPDA reductase, but unlike in plants, neither contains the peroxisome targeting sequence. Indeed, JA synthesis was not completely abolished in the absence of Opr1 function. It remains to be seen whether this is likely due to the compensatory role of Opr1-like and/or an alternative JA biosynthesis pathway.

In *M. oryzae*, GFP-Opr1 localised to the cytoplasm of vegetative mycelia, aerial hyphae, and basal and middle cells of the conidia. However, interestingly, the terminal cells of the conidia showed punctate localisation of GFP-Opr1, especially during germination. This suggests a spatiotemporal regulation of Opr1 expression, likely during the onset of polarised growth. Indeed, we observed a similar punctate GFP-Opr1 localisation pattern in the mature appressoria ready to form penetration pegs. Future studies will address whether such punctate GFP-Opr1 structures represent the late-endosomal vesicles that scaffold cyclic AMP-PKA signalling in *M. oryzae* [39].

We found that the exogenously added JA inhibits germ tube elongation and rather triggers appressorial development in *M. oryzae*. Interestingly, JA was found to share the signalling pathway with cAMP to activate the appressorial development. Further, our results indicate that JA acts just upstream of the MAPKKK Mst11. Importantly, the exogenously added JA only reduced the germ tube growth and did not induce appressorial development on a non-inductive soft hydrophilic surface. This strongly suggests that JA acts upstream of cAMP and MAPKKK to induce the switchover from polarised to isotropic growth during pathogenic development in *M. oryzae*. We further found that jasmonates such as 12-OPDA, JA and Me-JA, but not linolenic acid, could induce appressorium formation in the cAMP-signalling defective mutants, suggesting a significant crosstalk between the two signalling moieties. Recently, it was shown that a fungal oxylipin 5,8-diHODE, from *Aspergillus fumigatus*, has the ability to induce appressorial development in *M. oryzae* in an in vitro assay [51]. Constitutively activated PKA in the *mac1*-suppressor mutant (*mac1*D *sum1-99*) of *M. oryzae*, showed increased accumulation of oxilipins (10R)-HPODE and (5S,8R)-DiHODE, likely via the 7,8-Linoleate Diol Synthase (LDS; Jerneren et al., 2010). However, the LDS (*MGG_13239* and *MGG_10859*) gene-deletion mutants in *Magnaporthe* did not show any defects in the vegetative or pathogenic growth [52,53]. It would be important to gain deeper insights into the seemingly complex mechanism of oxylipin/jasmonate-based regulation of appressorial development in *Magnaporthe*. Overall, we have shown that *M. oryzae* Opr1 converts endogenous 12-OPDA to synthesize JA, which has a novel function in the cessation of polarised growth to control the germ tube growth and in cell signalling in concert with cAMP and ROS to induce pathogenic development in the blast fungus.

In conclusion, our study adds to the growing functional importance of oxygenated lipids in fungal biology [51], and provides new information and insights in implicating fungal jasmonate as a key factor that determines the timely cessation of germ tube growth and precise induction of appressorium formation; and suggests novel cyclic AMP-dependent control points that regulate JA signalling in response to intracellular oxidation in the rice blast pathogen. Future experiments will determine the exact intersection points and mechanistic cooperativity between jasmonate and cAMP signalling; and precise biosensor-based real time analysis of fungal JA to reveal novel/critical targets for controlling the devastating blast disease in rice and other important cereal crops.

## Figures and Tables

**Figure 1 jof-07-00693-f001:**
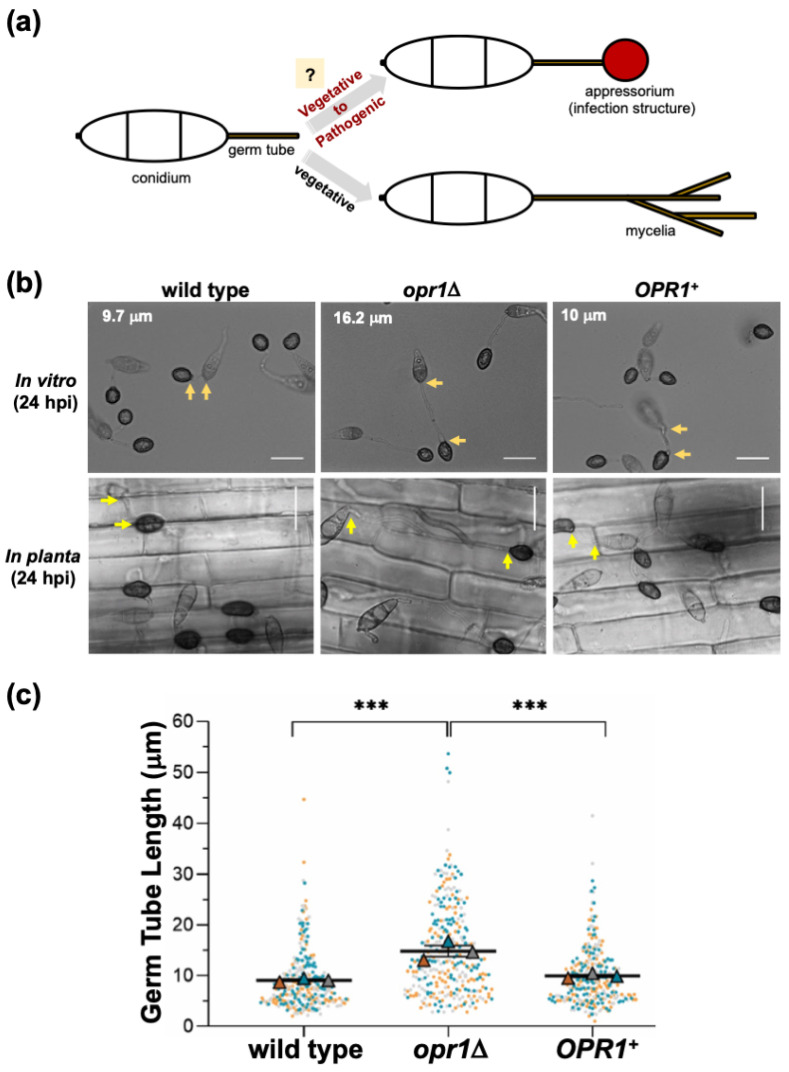
Loss of OPR1 results in extended germ tube growth prior to appressorium initiation in *M. oryzae*. (**a**) Schematic representation of the critical steps involved in infection-related morphogenesis in conidial germ tubes of *M. oryz*ae in response to inductive host environmental cues. The absence of such host factors causes cessation of this developmental switch and continuation of vegetative growth as mycelia without the formation of appressoria. (**b**) Bright-field micrographs showing comparative analysis of germ tube growth and appressorium formation in conidia from wild-type, *opr1*Δ, or the complemented *opr1*Δ (*OPR1^+^*) *M. oryzae* strains inoculated on artificial inductive surface (in vitro) or rice sheath (in planta). Compared to the WT or Complemented strain, germ tubes in *opr1*Δ mutant were significantly longer. Arrows demarcate the germ tube length in each panel. Scale bar equals 10 μm. (**c**) Graphical representation of the quantification of the germ tube length in the indicated *M. oryzae* strains. Each dot represents the individual data derived from three independent biological repeat experiments (*n* = 300 conidia for each sample). The overlapping triangles represent the means of each repeat in line with dots of the same colour. Two-tailed *t*-test was applied for the comparisons, *** *p* <0.001. Fresh conidia were inoculated on hydrophobic cover glass or rice sheath, and measurements were performed at 24 hpi.

**Figure 2 jof-07-00693-f002:**
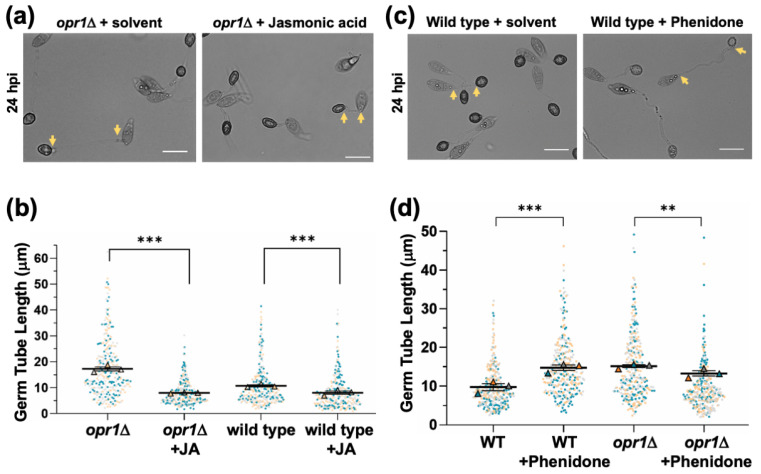
Exogenous JA suppresses germ tube defects in the opr1Δ mutant. (**a**) Conidia from the *opr1*Δ strain were inoculated on inductive surface in the presence (0.2 mM) or absence (mock/solvent control) of exogenous jasmonic acid, and germ tube and appressorial development were assessed and quantified at 24 hpi. Germ tube length was indicated by 2 arrows in each panel. Scale bar = 10 µm. (**b**) Exogenous JA significantly represses germ tube growth in *opr1*Δ and WT strain. Data represents mean ± SE from 3 independent repeats of the experiment, each involving at least 100 conidia. Each dot represents the value in the data from three biologically independent repeats (*n* = 300). Two-tailed *t*-test was performed to test the differences among the samples, ** *p* <0.01; *** *p* <0.001. (**c**,**d**) Chemical inhibition of JA synthesis (phenidone) results in elongated germ tubes and recapitulates the *opr1*Δ phenotype in the wild-type *M. oryzae* strain. Conidia from the WT strain were inoculated on inductive surface in the presence or absence of the JA-inhibitor Phenidone, and germ tube and appressorial development quantified at 24 hpi. Germ tube length was delineated by 2 arrows in each panel. Scale bar = 10 µm. Quantification performed as detailed for panel b above.

**Figure 3 jof-07-00693-f003:**
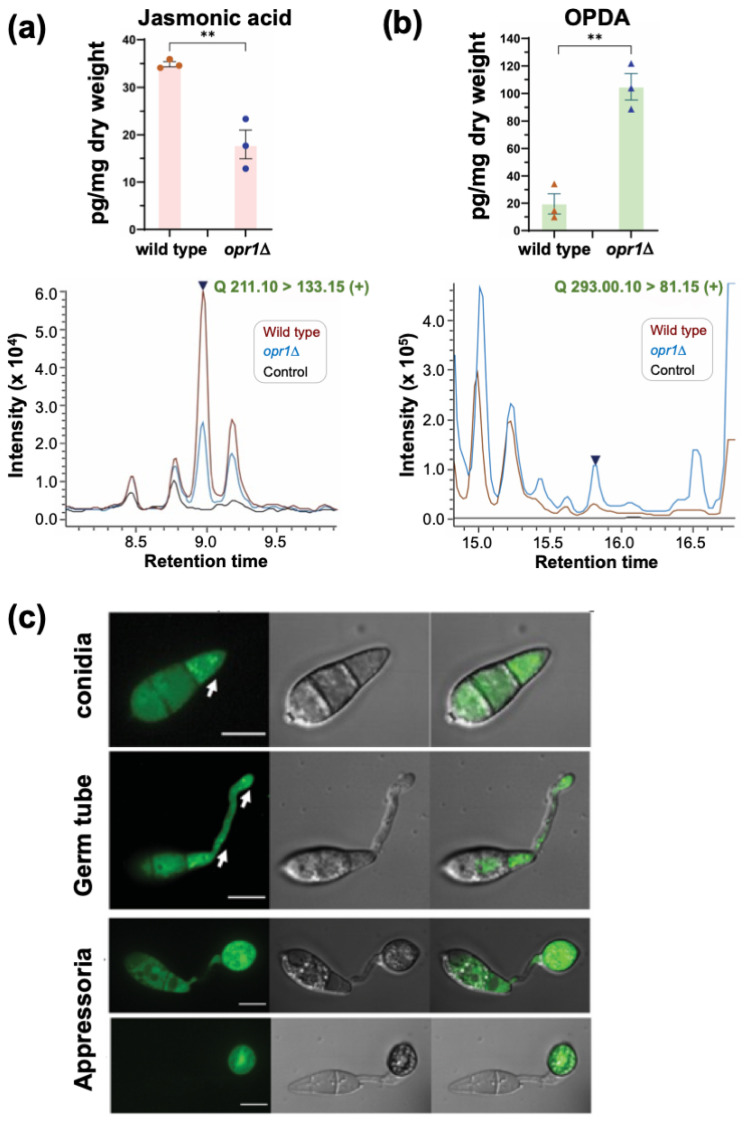
Opr1 is necessary for jasmonate biosynthesis and localizes to a distinct vesicular compartment during pathogenesis in *M. oryzae*. (**a**) JA production is halved in the oxylipin signature of *opr1*Δ compared to the wild type. Selective ion monitoring for the ion *m/z* = 211.10 was utilised for detection of fungal JA in wild-type (WT) and *opr1*Δ. Data are the mean ± SE of three repeats of the experiment. (**b**) A significantly higher accumulation of the substrate of Opr1, cis-OPDA, occurs in the *opr1*Δ, while only trace levels of the cis-OPDA are evident in WT extracts. (** *p* < 0.01; two-sided *t*-text). The dark blue peak in each spectrum represents the chemical standard used for JA or cis-OPDA, respectively. Please refer to the Methods section for more details. (**c**) Subcellular localization of GFP-Opr1 during pathogenic differentiation in *M. oryzae*. Conidia from the *GFP-OPR1* strain were inoculated on the inductive surface and imaged using confocal microscopy at the developing germ tube, incipient appressorium, and appressorial maturity stages of pathogenesis. Micrographs represent the confocal image for GFP-Opr1 (**left**), brightfield (**middle**) and the merged (**right**) at the indicated stage of development in *M. oryzae*. Arrows indicate the punctate/vesicular localization of GFP-Opr1. Scale bar = 5 μm.

**Figure 4 jof-07-00693-f004:**
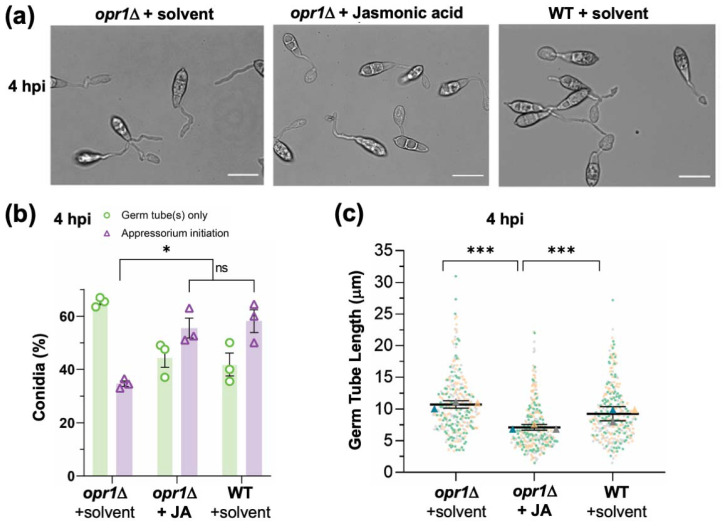
Loss of Opr1 significantly delays appressorium initiation in *M. oryzae*. (**a**) Exogenous JA corrects the delayed appressorium initiation defect in *opr1*Δ. Bright-field micrographs of the developing appressoria in *opr1*Δ and WT strain. Compared to the WT control, overall hooked structures (indicating the initiation of appressoria formation) in *opr1*Δ germ tubes were significantly reduced; but are restored upon treatment with exogenous JA. Conidia were inoculated and grown on the inductive cover glass surface for 4 h. Scale bar 10 μm. (**b**) Quantification of the chemical complementation experiment depicted in A. Data shown represents the mean ± SEM from three independent experiments, and two-tailed unpaired Student’s *t*-test determined *p*-values, * *p* < 0.05. *n* = 300 conidia per experiment (**c**) Dots and box plot depicting germ tube length in conidia that have initiated appressorium formation in opr1Δ and WT strain of *M. oryzae*. Data (*n* = 300) is derived from three independent experiments, with two-tailed *t*-test performed to evaluate the significance values, *** *p* < 0.001.

**Figure 5 jof-07-00693-f005:**
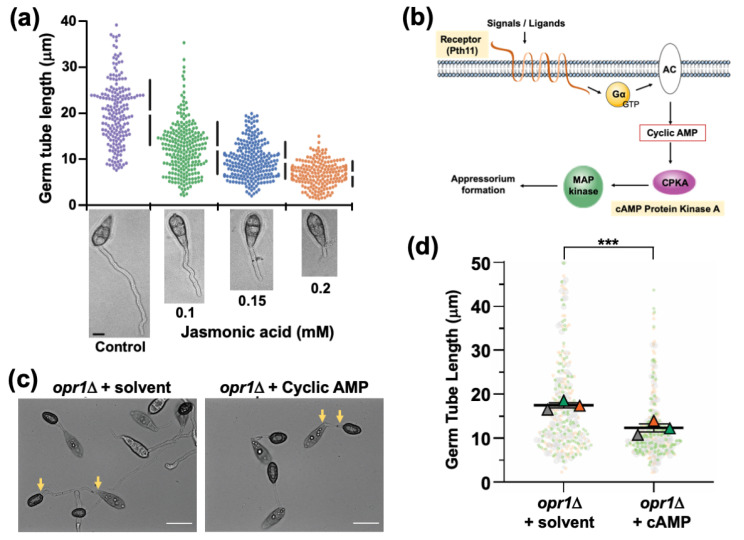
JA regulates germ tube development and appressorium formation in concert with the cyclic AMP signaling in Magnaporthe. (**a**) Exogenous JA shortens the germ tube length/growth on a non-inductive surface. Conidia from the WT strain were inoculated on a non-inductive hydrophilic surface in the presence of the indicated amounts of JA and germ tube growth assessed at 8 hpi. Data represents mean ± SE from three independent repeats of the experiment, each involving at least 300 conidia. Micrographs of the WT germ tubes formed on a non-inductive surface upon treatment with the indicated JA concentration. Scale bar equals 5 μm. (**b**) Schematic depiction of the heterotrimeric G proteins/cyclic AMP-PKA signalling cascade essential for appressorium formation in the rice blast fungus. Exogenous stimuli activate the membrane-localized GPCR Pth11 to trigger the downstream G proteins. Mac1, an adenylate cyclase, catalyses the ATP to cAMP conversion. Induced cAMP during pathogenic differentiation activates the CPKA kinase, triggering the downstream signaling, including the Pmk1-MAPK cascade, thus leading to appressorium formation. (**c**) Exogenous cAMP suppresses the dual defects of elongated germ tubes and delayed appressorium formation in the opr1Δ mutant. Germ tube development and appressorium formation was assessed on the inductive surface in opr1Δ conidia in the presence or absence of cyclic AMP. Images were captured at 24 hpi. Scale bar = 10 µm. (**d**) Graphical depiction of the quantitative analysis of the germ tube length in conidia, capable of appressorium formation in opr1Δ. Dots represent length measurement of each germ tube in three independent replicates of the experiment with *n* = 300 conidia in each instance. Triangles depict mean ± SE value of the three repeats. *** *p* < 0.001.

**Figure 6 jof-07-00693-f006:**
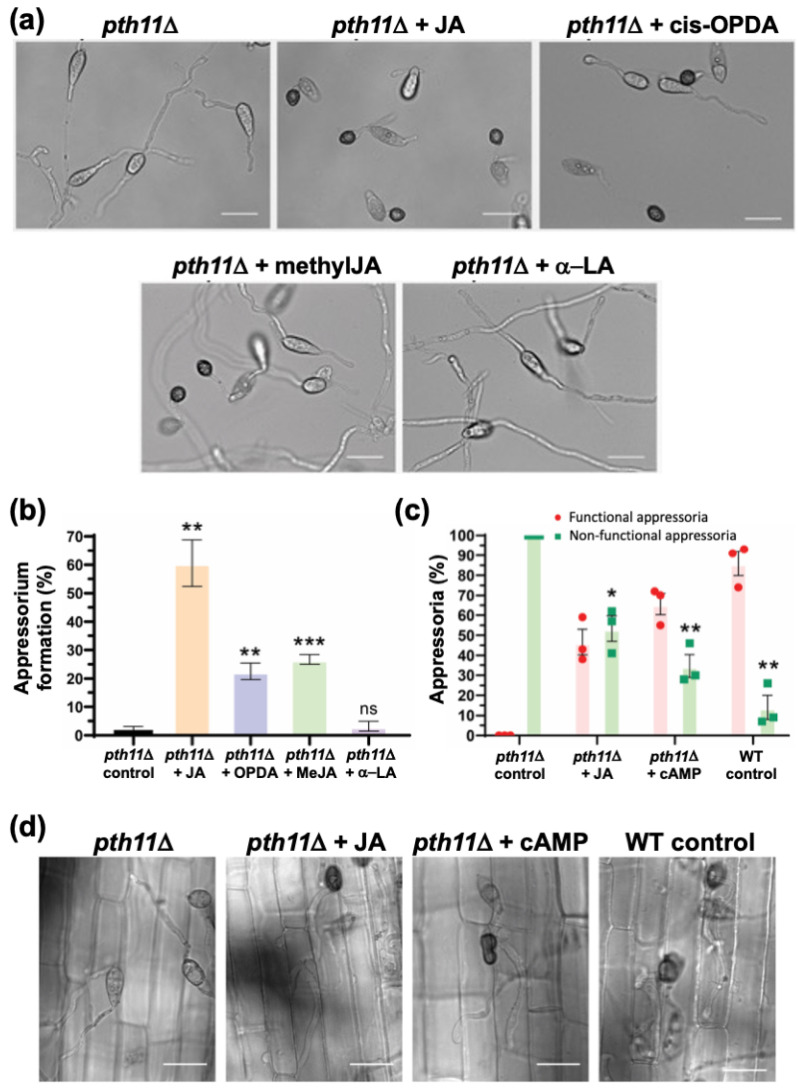
Jasmonate, cis-OPDA and MeJA, but not α-LA, restore appressorium formation and pathogenicity in the *pth11*Δ mutant. (**a**) Exogenous jasmonic acid restores appressorium formation in the *pth11*Δ mutant. Conidia from *pth11*Δ strain were inoculated on the inductive surface for 24 h in the absence or presence of the indicated oxylipin metabolite or a-linolenic acid. Treatment with JA, cis-OPDA or MeJA, but not a-LA significantly restored appressorium formation in *pth11*Δ. (**b**) Bar graph depicting the quantification of appressorium formation in the *pth11*Δ conidia upon treatment with the indicated oxylipin or precursor fatty acid. Solvent treated pth11Δ served as a negative control. (**c**) Oxylipins suppress *pth11*Δ defects in pathogenesis. Bar graph shows that the resultant appressoria in the oxylipin-treated *pth11*Δ mutant are functional, as they were able penetrate the host cells and generate the invasive hyphae at 28 hpi. Treatment with cAMP in parallel or with WT conidia served as positive controls. * *p* < 0.05; ** *p* < 0.01; *** *p* < 0.001 (unpaired two-tailed *t*-test; ns Not significant; *n* = 3 experiments). (**d**) Micrographs showing pathogenic development at 28 hpi in the indicated *M. oryzae* strains. The untreated *pth11*Δ control, which fails to form appressoria or invasive hyphae on rice sheath, served as a negative control. Scale bars represent 10 micron. Data represent mean ± SE from three biological replicates of the experiment using 300 conidia per sample in each instance.

**Figure 7 jof-07-00693-f007:**
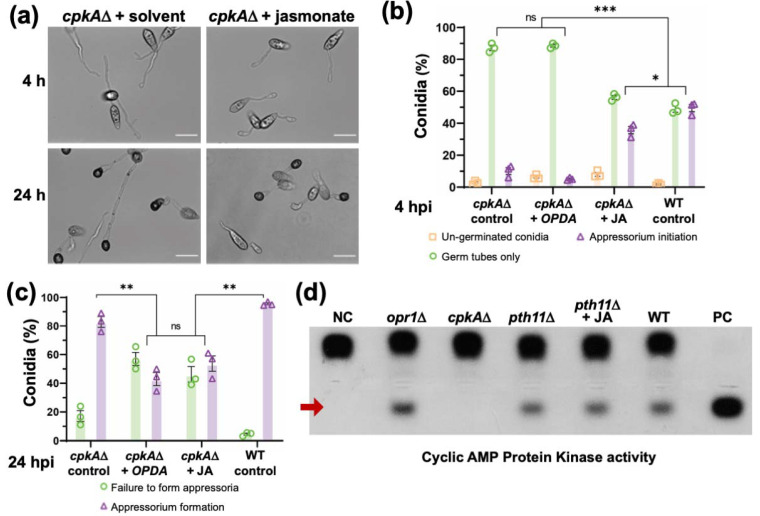
JA signaling functionally interconnects with the cAMP cascade at the CPKA node. (**a**) Exogenous JA, but not cis-OPDA, suppresses only the germ tube growth defect in the *cpkA*Δ mutant. Conidia from the *cpkA*Δ strain were treated with JA or cis-OPDA and germ tube development and appressorium initiation and formation were analysed at 4 hpi and 24 hpi, respectively. Scale bar equals 10 micron. (**b**,**c**) JA or cis-OPDA fail to restore proper appressorium development in *cpkA*Δ. Bar graphs representing quantitative data analysis for the experiment described in (**a**). Although the oxylipin-treated germ tubes were shorter in length, the percentage of appressorium formation in such JA or cis-OPDA treated *cpkA*Δ conidia was significantly reduced. Data represent mean ± SE from three biological replicates each using 300 conidia as the sample size. * *p* < 0.05; ** *p* < 0.01; *** *p* < 0.001 (unpaired two-tailed *t*-test; ns Not significant; *n* = 3 experiments). (**d**) Loss of OPR1 or the supplementation with JA does not affect the intrinsic cAMP-Dependent Protein Kinase A (PKA) activity. Quantification of the intracellular cAMP PKA activity in total extracts from the indicated wild-type and mutant *M. oryzae* strains was performed using the non-radioactive CPKA assay system and the model substrate, Kemptide.

## Data Availability

The data that support the findings of this study are available from the corresponding author upon reasonable request. Protein sequence for Opr1 is available at Uniprot and Ensembl (Available online: https://fungi.ensembl.org/Magnaporthe_oryzae/Info/Index (accessed on 4 May 2021).) under the locus ID: MGG_10583.

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
