# Peer review of "Fungal Jasmonate as a Novel Morphogenetic Signal for Pathogenesis"

_jof, 2021, doi:10.3390/jof7090693_

Round 1

Reviewer 1 Report

This is a well written manuscript that presents worthwhile information and was carried out with appropriate controls. I have a few very minor suggestions for improvement.

Figure 3C legend should explain what the 3 different panels/columns represent. They appear to be confocal > bright field > merge. That needs to be better described.

Line 390 has a few words that are italicized that should not be.

Line 505 - Need to define the term ROS here at first mention of the acronym.

Reference 53 is missing the full citation.

Author Response

Comments and Suggestions for Authors:

This is a well written manuscript that presents worthwhile information and was carried out with appropriate controls. I have a few very minor suggestions for improvement.

Author Response: Many thanks for the positive response to our manuscript, and for considering it important to the field of fungal cell biology and pathobiology.

Figure 3C legend should explain what the 3 different panels/columns represent. They appear to be confocal > bright field > merge. That needs to be better described.

Author Response: Many thanks for pointing this out. As suggested, we have now included a description for each panel shown in the Figure 3C legend. The sentence reads: Micrographs represent confocal image for GFP-Opr1 (left), brightfield (middle) and the merged (right) at the indicated stage of development in M. oryzae.

Line 390 has a few words that are italicized that should not be.

Author Response: Many thanks. The italicized words have now been changed to regular font.

Line 505 - Need to define the term ROS here at first mention of the acronym.

Reference 53 is missing the full citation.

Author Response: ROS has now been changed to “reactive oxygen species” in sentence 505.

The full citation for the dissertation cited in Reference 53 has now been provided.

Reviewer 2 Report

  This is an excellent research work. It opened the way to further in-depth understanding of the the penetration peg formation and further steps in the disease process.

Author Response

Comments and Suggestions for Authors:

This is an excellent research work. It opened the way to further in-depth understanding of the penetration peg formation and further steps in the disease process.

Author Response: Many thanks for the affirmative and expedited response to our study, and for finding it to be significant in providing an in-depth molecular understanding of the initiation of blast disease in rice.